# Serum presepsin level reflects macrophage activation and hemophagocytosis in bone marrow

Katsumasa Yamanaka[1], Kazuki Furuhashi [2]*, Kumiko Shimoyama[3], Yusuke Okubo[3], Toshitaka Yukishima [3], Saori Sawada[1], Miwa Adachi[3], Keita Yamashita[4], Moriya Iwaizumi[4], Yasuyuki Nagata[3], Takaaki Ono[3], Noriyoshi Ogawa[3], Masato Maekawa[4], Satoshi Baba[1]

1 Department of Diagnostic Pathology, Hamamatsu University School of Medicine, Hamamatsu, Japan, 2 Infection Control and Prevention Center, Hamamatsu University Hospital, Hamamatsu, Japan, 3 Third Division, Department of Internal Medicine, Hamamatsu University School of Medicine, Hamamatsu, Japan, 4 Department of Laboratory Medicine, Hamamatsu University School of Medicine, Hamamatsus, Japan

* k.furu@hama-med.ac.jp

## Abstract

### Background

Presepsin (soluble CD14 subtype) is a clinically used diagnostic biomarker for sepsis. It is secreted into the blood by activated macrophages, and serum concentrations are elevated in patients with sepsis. Previous reports suggest that presepsin may be secreted into the peripheral blood due to increased hemophagocytosis. However, the relationship between serum presepsin concentration and hemophagocytosis, as evidenced by bone marrow aspiration for diagnosing hemophagocytic lymphohistio-cytosis, is not well understood. Therefore, we investigated the relationship between serum presepsin level and hemophagocytes in bone marrow.

### Methods

We retrospectively analyzed 61 patients with serum presepsin level who underwent bone marrow aspirations. We examined the relationship between serum presepsin level and other laboratory findings, including the number of macrophages and the percentage of hemophagocytes in their bone marrow. Macrophages and hemophagocytes were counted using bone marrow smears. Immunostaining of bone marrow aspirate smears was performed using a CD14 antibody to evaluate the relationship between serum presepsin level and hemophagocytes in the bone marrow.

### Results

Serum presepsin level correlated with inflammatory markers (C-reactive protein and D-dimer) and markers related to hemophagocytosis (lactate dehydrogenase, ferritin, red blood cell count, hemoglobin, hematocrit, number of macrophages, and

**Data availability statement:** The data underlying the results presented in the study are available from the Hamamatsu University School of Medicine, Research Collaboration Division, Research Collaboration Section. 1-20-1, Handayama, Chuo-ku, Hamamatsu-city, Shizuoka, Japan TEL: +81-53-435-2680 E-mail: kenkyou.k@hama-med.ac.jp.

**Funding:** The author(s) received no specific funding for this work.

**Competing interests:** The authors have declared that no competing interests exist.

percentage of hemophagocytes in the bone marrow). The percentage of hemophagocytes in the bone marrow was positively correlated with serum presepsin level ($r = 0.435$, $p < 0.001$) and with ferritin ($r = 0.438$, $p = 0.015$), both of which were elevated during hemophagocytosis. CD14 expression is attenuated in hemophagocytes in bone marrow and lymph nodes.

## Conclusion

These findings suggest that serum presepsin is released by hemophagocytes and reflects the activation of macrophages and hemophagocytosis in bone marrow.

---

## Introduction

In vivo hemophagocytosis by monocytes/macrophages is primarily caused by the removal of old blood cells. Hemophagocytic lymphohistiocytosis (HLH), a pathological condition caused by excessive hemophagocytosis, can result from genetic factors or induced by infection, malignancy, or autoimmune disease [1–4]. Clinically, the disease is characterized by high fever, cytopaenia, and splenomegaly. Laboratory findings include cytopenia, elevated levels of transaminases, lactate dehydrogenase (LD), soluble interleukin-2 receptor (sIL-2R), and ferritin; and bone marrow examination often reveals hemophagocytosis by macrophages [5–8]. Presepsin (soluble CD14 subtype) has recently emerged as a valuable diagnostic and prognostic marker for sepsis in emergency medicine. It is a truncated N-terminal fragment of CD14 expressed on the membrane surface of monocytes/macrophages with a molecular weight of 13 kDa [9,10]. When monocytes/macrophages phagocytose bacteria, CD14 expressed on the surface of the phagocytic cells is taken up by the bacteria and degraded intracellularly to be produced and released as presepsin [11–13]. Even in healthy individuals, exposure to microorganisms results in a small amount of presepsin being released into the blood [14]. Serum presepsin level is markedly increased in patients with systemic infections, indicating strong specificity. Furthermore, the short half-life of presepsin makes it useful for the early detection of sepsis [10,15,16]. However, it has been suggested that presepsin is secreted into the peripheral blood and may increase due to the phagocytosis of blood cells. Ikegami et al. reported elevated presepsin level in patients with HLH and elucidated the mechanism by which monocytes/macrophages phagocytose neutrophil extracellular traps and produce presepsin [17]. The relationship between HLH and presepsin remains incompletely understood. While bone marrow aspiration is performed to diagnose HLH, the relationship between the hemophagocytic activity observed in the bone marrow and serum presepsin level has not yet been fully elucidated. Therefore, this study analyzed the relationship between serum presepsin level and bone marrow monocytes/macrophages acting as hemophagocytes and evaluated their utility in diagnosing HLH.

## Materials and methods

### Participants

This study retrospectively evaluated 61 patients who underwent bone marrow aspiration between January and July, 2021. Patients with an estimated glomerular filtration rate (eGFR) <30 (mL/min/1.73 m$^2$) were excluded [18]. The diagnosis of HLH was based on the diagnostic criteria proposed by Imashuku S. [19]. In addition, Macrophage activation syndromes (MAS) was ruled out after consultation with two immunologists. This study was approved by the Institutional Review Board Committee of Hamamatsu University School of Medicine (21–191). Access to electronic medical records for research purposes occurred from December 2023 to February 2024. During data extraction, the designated data manager accessed direct identifiers solely for record linkage purposes. Subsequently, the analysis dataset was fully anonymized, and researchers could not access information that would identify individual patients.

### Measurement of serum presepsin levels

This study included 61 consecutive cases with residual serum samples available for presepsin measurement following bone marrow aspiration. Bone marrow aspiration was performed at a time point preceding therapeutic intervention for the underlying disease in most cases. Serum samples were collected from residual patient samples. Those were obtained using coagulation-promoting tubes manufactured by Sekisui Medical Co., Ltd. (Tokyo, Japan) and centrifuged at 3,500 rpm for 8 minutes, and the separated serum was stored at −80 °C until analysis. Serum presepsin level was measured using the HISCL™ Presepsin Kit (Sysmex Corporation, Kobe, Japan). The measurement range was 20–30,000 pg/mL, and the cutoff value for bacterial sepsis was 500 pg/mL.

### Calculation of the percentage of hemophagocytes in bone marrow smears

Macrophages and hemophagocytes in bone marrow were morphologically evaluated by two independent pathologists using bone marrow smears. Macrophages were counted in 100 low-power fields (×200) using microscopy. Hemophagocytes were defined by the presence of one or more blood cells in the cytoplasm of macrophages in bone marrow smears (Fig 1). The percentage of hemophagocytes was calculated as the number of hemophagocytic macrophages among 50 macrophages, and this value was also expressed per 100 macrophages (%) for clarity in data presentation. Based on reference data [20–23], a total nucleated cell count within the range of 10–25 × 10$^4$/μL can reasonably be regarded as normal. Furthermore, to clearly exclude incidental findings, this study operationally defined "increased hemophagocytes" as ≥ 1% of nucleated cells (i.e., ≥ 5 hemophagocytes per 500 nucleated cells counted).

### Evaluation of macrophage markers in bone marrow and lymph nodes by immunohistochemistry and hematoxylin and eosin staining

Bone marrow and lymph node samples from patients were paraffin-embedded and sectioned. Bone marrow samples were obtained from patients with B-cell lymphoma without bone marrow infiltration and from patients with HLH after bone marrow transplantation. Lymph node samples were collected from patients with lung cancer or B-cell lymphoma. Bone marrow and lymph node samples from the subject patients were fixed in 10% neutral-buffered formalin, dehydrated through a graded ethanol series, and embedded in paraffin. Paraffin-embedded tissues were cut into 3-μm sections and mounted onto glass slides. The sections were deparaffinized in xylene, rehydrated through graded alcohols, and then subjected to hematoxylin and eosin (HE) staining or immunohistochemistry (IHC). For IHC, antigen retrieval was performed in Tris-EDTA buffer (pH 9.0) at 95 °C for 40 minutes. After blocking endogenous peroxidase with 3% H$_2$O$_2$ for 5 minutes, the slides were incubated with the primary antibodies CD14 (clone: D7A2T, 1:200 dilution; Cell Signaling Technologies, Danvers, MA, USA) and CD68 (clone: PG-M1, 1:100 dilution; Dako, Glostrup, Denmark) for 30 minutes. Subsequently, the sections were incubated with visualization reagent (Histofine simple stain MAX-PO (MULTI); Nichirei Co., Tokyo, Japan)

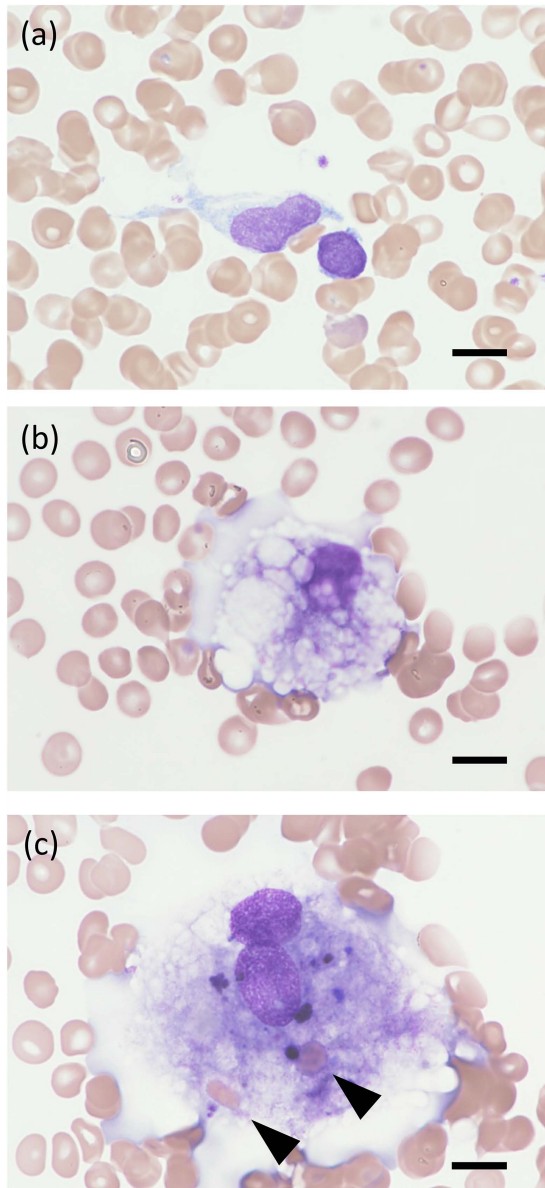

**Fig 1. Comparison of macrophages morphology in bone marrow.** (a) A representative image of non-activated macrophages and non-hemophagocytes. (b) A representative image of activated macrophages and non-hemophagocytes. (c) A representative image of activated macrophages and hemophagocytes. Arrowheads show phagocytized red blood cells, Scale bar = 10 μm.

for 30 minutes, and the immunoreaction was visualized using the Dako REAL EnVision Detection System, Peroxidase/DAB+, Rabbit/Mouse (Dako) and counterstained with hematoxylin.

## Statistical analysis

All statistical analyses were performed using GraphPad Prism software (version 9.0; GraphPad Software Inc., CA, USA). Data were expressed as mean ± standard deviation (SD) and median (Interquartile range: IQR). Differences between

groups were assessed using Spearman's rank correlation coefficient. Differences were considered statistically significant at $p < 0.05$.

## Ethics approval statement

This study was conducted in accordance with the Declaration of Helsinki and was approved by the Ethics Committee of Hamamatsu University School of Medicine (Approval No. 21–191).

## Patient consent statement

The need for informed consent was waived by the Ethics Committee of Hamamatsu University School of Medicine due to the retrospective nature of the study.

## Results

In this study, 61 patients were included. Patient demographics and clinical characteristics are shown in Table 1. The mean age was 58.5 years (range, 20–88 years). The primary diseases were mainly hematological malignancies (e.g., multiple myeloma, malignant lymphoma, acute myeloid leukemia, and myelodysplastic syndrome). Laboratory findings are shown in Table 2. The mean laboratory values were as follows: blood urea nitrogen (BUN), $16.3 \pm 8.3$ mg/dL; creatinine, $0.9 \pm 0.3$ mg/dL; estimated glomerular filtration rate eGFR, $68.3 \pm 23.0$ mL/min/1.73 m²; total bilirubin, $0.8 \pm 0.4$ mg/dL; triglycerides, $173.4 \pm 88.7$ mg/dL; aspartate aminotransferase (AST), $28.2 \pm 17.7$ U/L; C- reactive protein (CRP), $1.1 \pm 2.5$ mg/dL. Notably, the LD, ferritin, and presepsin levels were increased with their respective concentrations of $378.7 \pm 727.2$ U/L, $821.2 \pm 1579.2$ ng/dL, and $303.8 \pm 288.8$ pg/mL. Hematological parameters revealed elevated white blood cell counts $(13.6 \pm 55.2 \times 10^9/L)$ and decreased hemoglobin levels $(11.5 \pm 2.7$ g/dL). D-dimer as well as soluble fibrin levels were

**Table 1. Patients' demographic and clinical characteristics at the time of bone marrow aspiration.**

| Characteristics (n=61) | mean±SD | median (IQR) |
|---|---|---|
| median age, years | 58.5±16.6 | 63 (48-71) |
| gender (male/female), n | 35/ 26 | |
| height (cm) | 162.0±10.5 | 163.5 (156-169) |
| weight (kg) | 57.3±10.8 | 56.2 (48.5-64.1) |
| BMI (kg/m²) | 21.9±3.9 | 22.2 (19.5-28.2) |
| disease, n (%) | | |
| Hematological malignancy | | |
| Multiple myeloma | 16 (26.2) | |
| Malignant lymphoma | 12(19.7) | |
| Acute myeloid leukemia | 10 (16.4) | |
| Myelodysplastic syndrome | 6 (9.8) | |
| Chronic myeloid leukemia | 5 (8.2) | |
| Acute lymphoblastic leukemia | 4 (6.6) | |
| Others | 3 (4.9) | |
| Autoimmune disease | | |
| Systemic lupus erythematosus | 3 (4.9) | |
| Autoimmune hemolytic anemia | 1 (1.6) | |
| Idiopathic thrombocytopenic purpura | 1 (1.6) | |

BMI, body mass index; SD, standard deviation; IQR, Interquartile range.

**Table 2. Laboratory findings at the time of bone marrow aspiration.**

| Characteristics | mean±SD | median (IQR) |
|---|---|---|
| Clinical chemistry test | | |
| BUN (mg/dL) | 16.3±8.3 | 15.2 (12.5-19.1) |
| creatinine (mg/dL) | 0.9±0.3 | 0.84 (0.69-1.05) |
| eGFR (mL/min/1.73 m²) | 68.3±23.0 | 67 (54-87) |
| total bilirubin (mg/dL) | 0.8±0.4 | 0.7 (0.5-1.0) |
| direct bilirubin (mg/dL) | 0.2±0.1 | 0.2 (0.1-0.2) |
| indirect bilirubin (mg/dL) | 0.7±0.4 | 0.6 (0.4-0.9) |
| triglyceride (mg/dL) | 173.4±88.7 | 158 (105-221) |
| LD (U/L) | 378.7±727.2 | 222 (191-317) |
| AST (U/L) | 28.2±17.7 | 24 (20-30) |
| ALT (U/L) | 30.4±29.6 | 22 (17-32) |
| ALP (U/L) | 84.1±29.6 | 75 (64-102) |
| CRP (mg/dL) | 1.1±2.5 | 0.14 (0.07-0.8) |
| ferritin (ng/dL) | 821.2±1579.2 | 289 (92-678) |
| presepsin (pg/mL) | 303.8±288.8 | 222 (154-313) |
| Complete blood count | | |
| white blood cell (10⁹/L) | 13.6±55.2 | 4.17 (2.93-6.05) |
| neutrophil (%) | 56.8±16.9 | 56 (48-62) |
| red blood cell (10¹²/L) | 3.7±1.0 | 3.72 (3.03-4.35) |
| hemoglobin (g/dL) | 11.5±2.7 | 11.8 (9.3-13.3) |
| hematocrit (%) | 34.5±7.7 | 35.5 (27.9-39.8) |
| reticulocyte (%) | 3.3±5.7 | 1.9 (1.5-2.3) |
| platelet (10⁹/L) | 180.1±132.9 | 158 (100-242) |
| Coagulation test | | |
| PT-INR | 1.1±0.4 | 0.98 (0.90-1.09) |
| APTT (sec) | 30.2±6.0 | 29.6 (26.9-32.0) |
| fibrinogen (mg/dL) | 349.2±132.5 | 334 (278-398) |
| D-dimer (µg/mL) | 6.0±9.0 | 2.4 (1.3-5.0) |
| soluble fibrin (µg/mL) | 15.5±23.6 | 5.3 (3.1-15.9) |
| indings of born marrow aspiration | | |
| nucleated cell count in bone marrow (10⁹/L) | 77.8±182.3 | 37 (22-74) |
| number of macrophage in bone marrow (/100 LPF) | 16.1±21.1 | 10 (6-20) |
| percentage of hemophagocyte in bone marrow (%) | 7.5±12.7 | 2 (2-8) |

BUN, blood urea nitrogen; eGFR, estimated Glomerular Filtration Rate; LD, lactate dehydrogenase; AST, aspartate aminotransferase; ALT, alanine aminotransferase; ALP, alkaline phosphatase; CRP, C-reactive protein; PT-INR, prothrombin time-international normalized ratio; APTT, activated partial thromboplastin time; 100 LPF, 100 low-power fields of view; SD, standard deviation; IQR, Interquartile range.

elevated (6.0±9.0 µg/mL and 15.5±23.6 µg/mL, respectively). Bone marrow findings showed that the nucleated cell count (77.8±182.3×10⁹/L) was within the normal range and an elevated percentage of hemophagocytes (7.5±12.7%). In the 61 cases analyzed in our study, the frequency of each finding based on the HLH diagnostic criteria proposed by Imashuku et al. [19] is shown in Supplementary Table 1. Only one case met the criteria for a confirmed diagnosis of HLH. Anemia was present in 16.4% (10/ 61), thrombocytopenia in 24.6% (15/ 61), and neutropenia in 11.5% (7/ 61). At least two types of cytopenia were present in 14.8% (9/ 61).

## Serum presepsin level correlates the inflammatory and hemophagocytic markers

In this study, the mean serum presepsin level was 303.8±288.8 pg/mL, and that of its median was 222 pg/mL (IQR 154−313 pg/mL). Significant positive correlations were observed between direct bilirubin (r=0.627, p=0.019), LD (r=0.405, p=0.001), alkaline phosphatase (ALP) (r=0.254, p=0.048), CRP (r=0.488, p<0.001), ferritin (r=0.438, p=0.015), D-dimer (r=0.543, p=0.013), and prothrombin time-international normalized ratio (PT-INR) (r=0.389, p=0.008). Conversely, negative correlations were observed between presepsin and red blood cell count (r=−0.488, p<0.001), hemoglobin (r=−0.551, p<0.001), and hematocrit (r=−0.560, p<0.001) (Table 3). Additionally,

**Table 3. Correlation between serum presepsin levels and other biomarkers including bone marrow aspiration findings.**

|  | r | p value |
|---|---|---|
| Clinical laboratory test |  |  |
| BUN | 0.114 | 0.382 |
| creatinine | 0.133 | 0.306 |
| eGFR | −0.150 | 0.249 |
| total bilirubin | −0.131 | 0.313 |
| direct bilirubin | 0.627 | 0.019 |
| indirect bilirubin | −0.143 | 0.275 |
| triglyceride | −0.102 | 0.476 |
| LD | 0.405 | 0.001 |
| AST | 0.147 | 0.259 |
| ALT | 0.017 | 0.898 |
| ALP | 0.254 | 0.048 |
| CRP | 0.488 | <0.001 |
| ferritin | 0.438 | 0.015 |
| Complete blood count |  |  |
| white blood cell | 0.052 | 0.692 |
| neutrophil | −0.109 | 0.433 |
| red blood cell | −0.507 | <0.001 |
| hemoglobin | −0.551 | <0.001 |
| hematocrit | −0.560 | <0.001 |
| reticulocyte | 0.086 | 0.526 |
| platelet | −0.208 | 0.107 |
| Coagulation test |  |  |
| PT-INR | 0.389 | 0.008 |
| APTT | 0.009 | 0.955 |
| fibrinogen | 0.021 | 0.889 |
| D-dimer | 0.543 | 0.013 |
| soluble fibrin | 0.067 | 0.791 |
| Findings of born marrow aspiration |  |  |
| nucleated cell count in bone marrow | −0.144 | 0.287 |
| number of macrophage in bone marrow | 0.303 | 0.018 |
| percentage of hemophagocyte in bone marrow | 0.435 | <0.001 |

BUN, blood urea nitrogen; eGFR, estimated Glomerular Filtration Rate; LD, lactate dehydrogenase; AST, aspartate aminotransferase; ALT, alanine aminotransferase; ALP, alkaline phosphatase; CRP, C-reactive protein; PT-INR, prothrombin time-international normalized ratio; APTT, activated partial thromboplastin time.

presepsin level was significantly correlated with bone marrow findings, including the number of macrophages (r = 0.303, p = 0.018) (Fig 2a) and the percentage of hemophagocytes (r = 0.435, p < 0.001) (Fig 2b).

**Presepsin precursor CD14 protein expression is attenuated in hemophagocytes in lymph nodes and bone marrow**

In normal lymph nodes, most macrophages are localized in the sinus and exhibit typical morphology. Both CD14 and CD68 are strongly expressed in these macrophages. (Fig 3 a-c). In contrast, sporadically activated macrophages were observed in the lymph nodes with hemophagocytosis, engulfing blood cells within their cytoplasm. These hemophagocytes showed strong CD68 expression, but decreased CD14 expression (Fig 3 d-f). In the normal bone marrow, monocytes and macrophages were scattered and appeared morphologically normal. CD14 and CD68 expression levels were also within the normal ranges (Fig 3 g-i). However, in bone marrow with hemophagocytosis, blood cells numbers were reduced, while macrophage numbers were increased. Activated macrophages exhibited prominent cytoplasmic vacuolization. Similar to the findings in the lymph nodes with hemophagocytosis, bone marrow hemophagocytes showed high CD68 expression, and markedly suppressed CD14 expression (Fig 3 j-l). Furthermore, comparison of macrophages in bone marrow before and after HLH onset revealed a decrease in CD14 expression on CD68 positive cells after HLH onset (S1 Fig e, f) compared to before HLH onset (S1 Fig b, c).

## Discussion

In this study, we have investigated the correlation between serum presepsin level and macrophage activation in the bone marrow, particularly during hemophagocytosis. While presepsin has been used as a marker for sepsis, it also correlates with markers of phagocytosis (such as direct bilirubin, LD, ferritin, and hemoglobin), as well as with CRP, an inflammatory marker. This suggests that presepsin could potentially be a novel marker for phagocytosis. CD14 is localized on the cell membranes of monocytes, macrophages, and neutrophils. Arai Y, et al. have reported that membrane form CD14 is internalized not only during bacterial phagocytosis but also during aseptic phagocytosis, such as that of sodium urate crystals. It is then cleaved by proteases like elastase and released as the soluble CD14, presepsin [11]. In this study, we demonstrated reduced CD14 expression on macrophages in a HLH case. We hypothesized that this reduction likely resulted from the same mechanism previously reported. Specifically, CD14 on the macrophage surface was internalized during phagocytosis of blood cells, cleaved by proteases, and consequently led to decreased the expression of CD14 on the cell surface.

Previous study has reported that in addition to elevated serum presepsin level in patients with HLH, the combination of serum presepsin level and other immune-related biomarkers is the useful predictor of treatment efficacy and mortality in patients with hematological malignancies and secondary HLH [24]. Presepsin level in sepsis patients, which is also commonly used in clinical practice, has been reported as 817.9 ± 572.7 pg/mL in sepsis patients and 1,992.9 ± 1,509.2 pg/mL in severe sepsis patients, compared to 294.2 ± 121.4 pg/mL in healthy individuals [9]. On the other hand, Nanno et al. reported that the median presepsin level at HLH onset in patients with hematological malignancies was 1,935 pg/mL (range 182–11,800 pg/mL) [24]. They suggested that HLH cases often showed elevated level comparable to those in severe sepsis, indicating a significant degree of macrophage activation. In our study, the presepsin level in a case meeting the HLH diagnostic criteria [19] was 1,895 pg/mL, comparable to previously reported values. Elevated presepin level in sepsis is primarily attributed to phagocytosis of bacterial components by monocytes, macrophages, and neutrophils [11]. In contrast, HLH is thought to be primarily caused by excessive macrophage activation and subsequent hemophagocytosis due to abnormal immune regulation triggered by infection or associated with underlying diseases such as malignant lymphoma or autoimmune disorders [24]. Therefore, we speculate that while presepsin elevation is observed in both sepsis and HLH, we speculate that in sepsis it may reflect the immune response to infection, whereas in HLH it may reflect the pathological overactivation of phagocytic cells such as macrophages in bone marrow and peripheral blood. However,

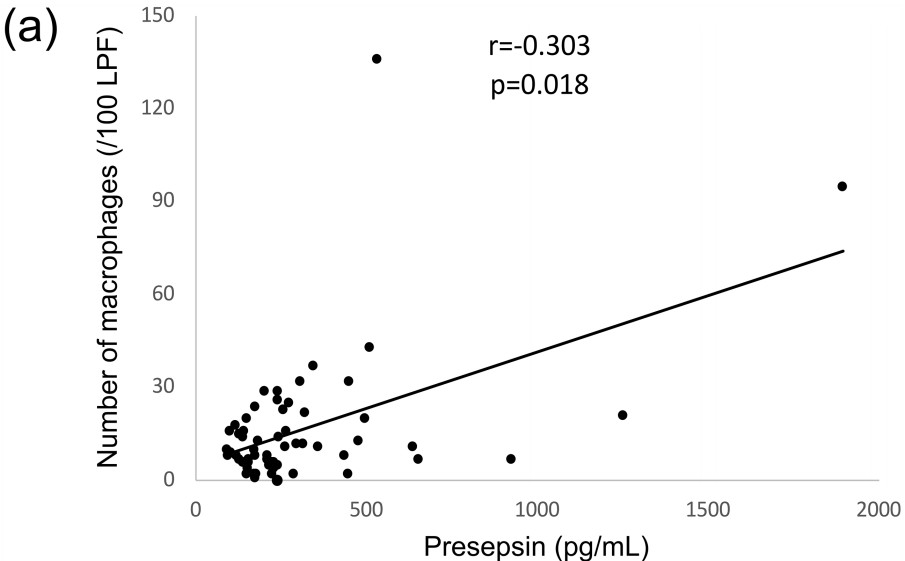

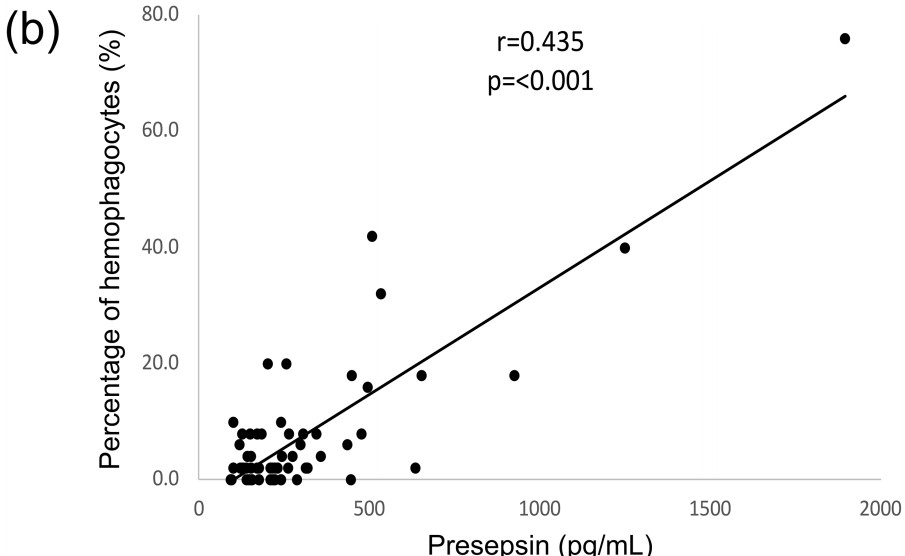

**Fig 2. Correlation between serum presepsin levels and bone marrow aspiration findings.** (a) Correlation with number of macrophages. (b) Correlation with number of hemophagocytes. The scores are calculated using Pearson's product-moment correlation coefficients. *p* < 0.05, 100 LPF: 100 fields of view at low power fields.

there have been no reports directly examining the relationship between serum presepsin level and hemophagocytosis using bone marrow tissue samples, as in the present study. Our results support these findings.

Furthermore, our results showed a positive correlation between serum presepsin level and the percentage of hemophagocytes. However, the correlation coefficient was not particularly high at 0.435. As HLH diagnosis cannot rely solely on serum presepsin level, combining serum presepsin level with other biomarkers is necessary to accurately detect HLH pathology. In children, it has been reported that serum IL-6, IL-10, IFN-γ, sIL-2R, and TNF-α levels are useful for diagnosing HLH [19,25–27]. In adults, serum ferritin level has been reported to be prognostic markers for HLH

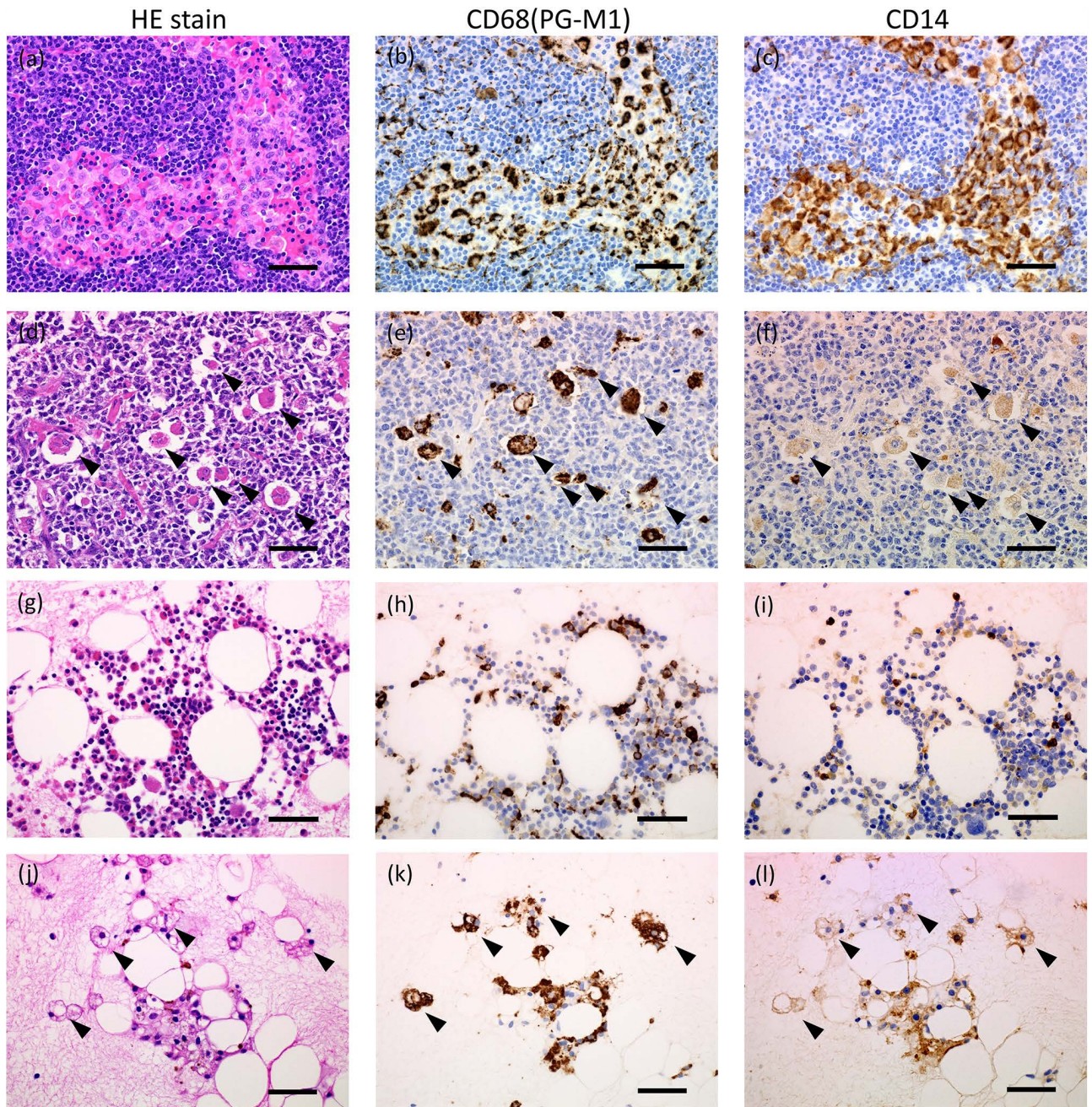

**Fig 3. Comparison of macrophage marker and presepsin precursor protein expression in the lymph node and bone marrow.** IHC is performed using CD68 (PG-M1) and CD14 (D7A2T) antibodies. The panels (a-c) show negative staining and (d-f) show positive staining for hemophagocytosis in the lymph node. Similarly, the panels (g-i) show negative staining and (j-l) show positive staining for hemophagocytosis in the bone marrow. Arrowheads show phagocytic cells. Scale bar = 50 μm.

[28,29] and serum adenosine deaminase level also increases in patients with HLH [30]. In addition, the combination of presepsin and sIL-2R reflecting macrophages and T cells activation, respectively, and may be a novel predictor of HLH severity and prognosis [24]. Despite our small sample size, we found a strong correlation between serum markers of

hemophagocytosis, β2-microglobulin (n = 21, r = 0.782, $p < 0.001$) and sIL-2R (n = 15, r = 0.689, $p = 0.006$), and the ratio of phagocytes in the bone marrow of participants in this study (result not indicated). Conversely, a study using monocytic cell lines showed increased CD14 in macrophages after phagocytosis [17]. This report evaluated CD14 expression *in vitro,* in macrophages after phagocytosis, at a very early stage immediately after phagocytosis. Most of our observed bone marrow findings were at a stage in which hemophagocytosis progressed during the clinical course. This suggests that CD14 expression in macrophages decreases as phagocytosis progresses. In addition, since the expression of CD68 on macrophages does not change during both steady state and hemophagocytosis, a decrease in the ratio of CD14 positive cells/CD68 positive cells in macrophages may be used as an indicator of hemophagocytes.

Serum presepsin level is affected by renal dysfunction and the blood samples agitation [18,31–33]. Although patients with severe renal dysfunction (eGFR < 30 mL/min/1.73 m$^2$) were excluded from this study, the inclusion of some patients with mild renal dysfunction, may have contributed to weak correlation. Additionally, we used the residual stored serum from patients for analysis. Therefore, the storage conditions and duration of serum supplementation may have influenced these results. Furthermore, patients who underwent blood tests and bone marrow aspiration during the study period were not necessarily evaluated at the time of diagnosis; some patients underwent bone marrow aspiration to monitor their condition after diagnosis, and others underwent bone marrow aspiration to observe their condition after the clinical diagnosis. This may have affected serum presepsin level. In future, we believe that by examining samples obtained through appropriate testing at the time of initial diagnosis in a prospective study, we will be able to more accurately clarify the relationship between serum presepsin level and hemophagocytes in the bone marrow.

The presence of hemophagocytes in the bone marrow is a critical finding for diagnosing HLH. However, as demonstrated in this study, hemophagocytes observed in bone marrow smears are not exclusively indicative of HLH, as they may occasionally be present in patients without HLH. Moreover, hemophagocytes are not always prominently identifiable in patients with HLH [6]. Therefore, relying solely on their presence in bone marrow smears for the diagnosis and differential diagnosis of HLH may result in relatively low diagnostic specificity [18]. Our findings suggest that serum presepsin level may reflect macrophage activation and hemophagocytosis in the bone marrow. While excluding sepsis and macrophage activation syndrome is essential when HLH is suspected, evaluating presepsin concentration in combination with existing biomarkers and clinical findings may provide non-invasive additional diagnostic value as a supplementary indicator of HLH pathophysiology. We believe that further accumulation of cases and investigation are necessary to clarify the extent to which serum presepsin measurement can contribute to HLH diagnosis and disease assessment.

## Supporting information

**S1 Fig. Comparison of macrophage marker and presepsin precursor protein expression in the bone marrow before and after the onset of HLH.** IHC was performed using CD68 (PG-M1) and CD14 (D7A2T) antibodies. Panels (a–c) show bone marrow findings before HLH onset: (a) hematoxylin and eosin (H&E) staining, (b) CD68, and (c) CD14. Panels (d–f) show bone marrow findings after HLH onset: (d) H&E, (e) CD68, and (f) CD14. Arrowheads indicate phagocytic cells. Scale bar = 50 μm.
(PDF)

**S1 Table. Frequencies of findings based on the diagnostic criteria for hemophagocytic lymphohistiocytosis.**
(DOCX)

## Acknowledgments

We thank Akihiro Masukawa (formerly of the Department of Laboratory Medicine, Hamamatsu University School of Medicine; currently at Beckman Coulter Inc.), Nao Tsurumi (Department of Laboratory Medicine, Hamamatsu University School of Medicine), and Satoshi Kiku (Sysmex Corporation) for their technical assistance.

## Author contributions

**Conceptualization:** Katsumasa Yamanaka, Kazuki Furuhashi, Kumiko Shimoyama.

**Data curation:** Katsumasa Yamanaka, Kazuki Furuhashi, Kumiko Shimoyama.

**Formal analysis:** Katsumasa Yamanaka, Kazuki Furuhashi, Kumiko Shimoyama, Yusuke Okubo, Saori Sawada.

**Investigation:** Katsumasa Yamanaka, Yusuke Okubo, Toshitaka Yukishima, Saori Sawada, Miwa Adachi, Keita Yamashita, Moriya Iwaizumi, Yasuyuki Nagata, Takaaki Ono, Satoshi Baba.

**Methodology:** Katsumasa Yamanaka, Kazuki Furuhashi, Miwa Adachi, Satoshi Baba.

**Project administration:** Katsumasa Yamanaka, Kazuki Furuhashi, Kumiko Shimoyama.

**Software:** Katsumasa Yamanaka, Kazuki Furuhashi.

**Supervision:** Noriyoshi Ogawa, Masato Maekawa, Satoshi Baba.

**Visualization:** Katsumasa Yamanaka, Kazuki Furuhashi.

**Writing – original draft:** Katsumasa Yamanaka, Kazuki Furuhashi.

**Writing – review & editing:** Katsumasa Yamanaka, Kazuki Furuhashi, Kumiko Shimoyama, Yusuke Okubo, Toshitaka Yukishima, Saori Sawada, Miwa Adachi, Keita Yamashita, Moriya Iwaizumi, Yasuyuki Nagata, Takaaki Ono, Noriyoshi Ogawa, Masato Maekawa, Satoshi Baba.

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
