## [Decision Letter · Decision Letter 0]

26 Sep 2025

Dear Dr.  Furuhashi,

Thank you for submitting your manuscript to PLOS ONE. After careful consideration, we feel that it has merit but does not fully meet PLOS ONE’s publication criteria as it currently stands. Therefore, we invite you to submit a revised version of the manuscript that addresses the points raised during the review process.

We look forward to receiving your revised manuscript.

Kind regards,

Zhaoqing Du, Ph.D

Academic Editor

PLOS ONE

Journal Requirements:

Reviewer's Responses to Questions

**Comments to the Author**

1. Is the manuscript technically sound, and do the data support the conclusions?

Reviewer #1: No

Reviewer #2: Partly

2. Has the statistical analysis been performed appropriately and rigorously?

Reviewer #1: No

Reviewer #2: N/A

3. Have the authors made all data underlying the findings in their manuscript fully available?

Reviewer #1: Yes

Reviewer #2: Yes

4. Is the manuscript presented in an intelligible fashion and written in standard English?

Reviewer #1: Yes

Reviewer #2: Yes

Reviewer #1: In this manuscript, Furuhashi et al. evaluated the significance of serum presepsin levels in 61 patients diagnosed with HLH. The authors reported positive correlations between presepsin and other inflammatory biomarkers and the presence of hemophagocytes in bone marrow samples.

Comments:

1. For presepsin to be clinically valuable in diagnosing HLH, it must improve the diagnostic performance beyond existing biomarkers (e.g., ferritin, sIL-2R, triglycerides). Specifically, it should enhance sensitivity and/or specificity, either by increasing the number of correct HLH diagnoses when combined with other markers or by reducing false positives. The authors should compare the diagnostic accuracy of presepsin with established HLH markers, both within the HLH cohort and in a comparator cohort (e.g., patients with sepsis or chemotherapy-induced pancytopenia).

2. Only a small proportion of patients in this cohort had histopathologic evidence of hemophagocytosis. How confident are the authors in their HLH diagnoses, as opposed to other macrophage activation syndromes?

3. At what time point were presepsin samples obtained, before or after initiation of treatment?

4. What is the mechanistic basis for elevated presepsin in HLH? The authors note that in sepsis, bacterial digestion of CD14 leads to its cleavage and subsequent cytokine release. In HLH, where red blood cells are phagocytosed, how does CD14 shedding occur from the macrophage surface?

5. What are the typical presepsin levels observed in sepsis, and how do they compare with those in HLH?

Reviewer #2: Reviewer comments to the author Date: 18 September, 2025

PONE-D-25-46766

Title: Serum presepsin levels reflect macrophage activation and hemophagocytosis in bone marrow

The manuscript submitted by Katsumasa Yamanaka et al. provides an important contribution to the scientific community and other concerned bodies regarding the correlation between Serum presepsin levels and hemophagocytosis in bone marrow. However, certain revisions are required to the submitted manuscript.

Major Comments

Abstract

Page 5, line 2: background is more preferable than objective, but include your objective under the heading of background.

Introduction:

1. Page 7, Line 16: space between the word “blood” and the reference.

Methods and materials

1. Participants is more formal than patients if possible.

2. How many of your study participants are with HLH

3. Page 9, line 4-5: Patients with an estimated glomerular filtration rate (eGFR) <30 (mL/min/1.73 m2) were excluded……. Needs citation (reference).

4. Page 9, line 14: Serum samples were collected from residual patient samples. What was the tube used to collect the blood, at what centrifugal force was the blood centrifuged to separate the serum? Please state this section in detail.

5. Bone marrow samples were obtained from patients with B-cell lymphoma without bone marrow infiltration and from patients with HLH after bone marrow transplantation. Lymph node samples were collected from patients with lung cancer or B cell lymphoma. Why you have collected bone marrow and lymph node samples only from those study participants since there are other hematological and autoimmune diseases included in your result section (table 1)? State clearly please?

6. You know that in histopathology lab, there are certain processes that the tissue section should undergo before mounting including tissue processing and staining. Thus, more detailed information is needed.

7. Page 11, Line 10: Data are expressed as mean ± standard deviation (SD). Are all your data continuous and normally distributed? Additionally, were is more preferable than are

Result

1. Page 1, Line 10-12: LD, Ferritin and presepsin are singular. Thus, say level, not levels and was, not were or you can state as “The LD, Ferritin and presepsin levels were increased with their respective concentration of …….

2. Page 12, Line 14: D-dimer levels were elevated at 6.0 ± 9.0 μg/mL, as well as soluble fibrin (15.5 15 ± 23.6 μg/mL). Rewrite it like D-dimer as well as soluble fibrin levels were elevated (6.0 ± 9.0 μg/mL and 15.5 15 ± 23.6 μg/mL respectively).

3. Page 12, Line 15-16: Bone marrow findings showed an increased nucleated cell count (77.8 ± 16 182.3×10⁹/L) and an elevated percentage of hemophagocytes (7.5 ± 12.7%). Define elevated nucleated cell and pathological percentage of hemophagocytes in bone marrow (i.e. need operational definitions).

4. Page 13, Line 3: In 61 patients, the median serum presepsin level was 311.2 pg/mL (range 91-1895 pg/mL). It is preferred to say “in this study, the median serum presepsin level was 311.2 pg/mL (range 91-1895 4 pg/mL). Additionally, in your method and material section under” Statistical analysis” you have stated that “Data are expressed as mean ± standard deviation (SD)”. Here you stated the serum presepsin level in median (range), but in mean± SD (303.8 ± 288.8) as shown in table 2 and page 12, line 12 (check your analysis again please).

5. Page 13, Line 10: Additionally, presepsin level was significantly correlated with bone marrow findings, including……, Not Additionally, presepsin levels significantly correlated with bone marrow findings, including……

6. Page13, Line 15-16: Presepsin precursor CD14 protein expression is attenuated in hemophagocytes in lymph nodes and bone marrow. No need of full stop (.) at the end since it is a sub-title.

7. Page 14, Line 4-6: In the normal bone marrow……………………..within the normal ranges. Support it with figure cross reference (fig…..)

8. Page14, Line 6-7: However, in bone marrow with hemophagocytosis, blood cells numbers were reduced, while macrophage numbers were increased. Please determine what proportion of your study participants have reduced RBC, WBC (leukopenia) and platelet count (thrombocytopenia) since HLH is characterized by cytopenia….. (Introduction… line 6)

Discussion

1. Page 15, Line 2: “In this study, we investigated serum and bone marrow smear specimens from 61 patients and found a correlation between serum presepsin levels and macrophage activation in the bone marrow, particularly during hemophagocytosis”. Paraphrase it again like” In this study, we have investigated the correlation between serum presepsin level and macrophage activation in the bone marrow, particularly during hemophagocytosis”.

2. Page 15, Line 17: Previous studies……. Implies more than one study, but you have cited only one study (18).

3. Page 16, line 15-17: “Despite our small sample size, we found a strong correlation between serum markers of 16 hemophagocytosis, β2-microglobulin (n=21, r=0.782, p<0.001) and sIL-2R (n=15, 17 r=0.689, p=0.006), and the ratio of phagocytes in the bone marrow of patients with HLH”. Please cite it if you have published it or state as “result not indicated”.

4 .page 16, 5-8: “In addition, since the expression of CD68 on macrophages does not change during both steady state and hemophagocytosis, a decrease in the ratio of CD14 positive cells/CD68 positive cells in macrophages may be used as an indicator of hemophagocytes”. Therefore, if possible, include the ratio of CD14+/CD68+ cells before and after hemophagocytosis in your result section.

Finally, conclude your study standing from your finding and forward your recommendations to the concerned body especially for those diagnosing HLH solely on hemophagocytes

Minor comments

1. Check your grammar.

Good luck!

**Do you want your identity to be public for this peer review?** For information about this choice, including consent withdrawal, please see our Privacy Policy

Reviewer #1: No

Reviewer #2: **Yes:** Sintayehu Admas

---

## [Author Response · Author response to Decision Letter 1]

11 Nov 2025

RESPONSE TO REVIEWER #1

1. For presepsin to be clinically valuable in diagnosing HLH, it must improve the diagnostic performance beyond existing biomarkers (e.g., ferritin, sIL-2R, triglycerides). Specifically, it should enhance sensitivity and/or specificity, either by increasing the number of correct HLH diagnoses when combined with other markers or by reducing false positives. The authors should compare the diagnostic accuracy of presepsin with established HLH markers, both within the HLH cohort and in a comparator cohort (e.g., patients with sepsis or chemotherapy-induced pancytopenia).

Thank you for your valuable suggestion. The aim of this study was to investigate the relationship between serum presepsin levels and hemophagocytosis in the bone marrow, it was not intended to directly verify the diagnostic accuracy of HLH. In this study, 61 consecutive patients who underwent both bone marrow aspiration and serum presepsin measurement at the same time were included, not all cases were diagnosed with HLH. Among them, only one case was finally diagnosed with HLH, and the remaining 60 cases did not meet the diagnostic criteria for HLH [Imashuku S. Int J Hamatol. 1997; 66: 135-151.]. Therefore, it was difficult to compare the diagnostic performance of presepsin with existing HLH markers such as ferritin, sIL-2R, and triglycerides in the data from this study. We agree with the reviewer's suggestion. As a future direction, we plan to include HLH cases in the target diseases for additional analysis and evaluate whether presepsin can contribute to improving diagnostic accuracy when compared with or combined with existing HLH biomarkers. We apologize for the insufficient expression in the main text. Supplementary Table 1 presents the frequency of findings based on the HLH diagnostic criteria analyzed in this study. Furthermore, we included a description of one case meeting the diagnostic criteria in the Materials and Methods and the Results section.

2. Only a small proportion of patients in this cohort had histopathologic evidence of hemophagocytosis. How confident are the authors in their HLH diagnoses, as opposed to other macrophage activation syndromes?

Thank you for your very important comments. As we responded to the previous reviewer’s comment, this study aimed to evaluate the association between serum presepsin levels and hemophagocytosis in the bone marrow. Therefore, we included only one case of HLH and described the characteristics of the study subjects in Supplementary Table 1. As the reviewer pointed out, we recognize the difficulty in differentiating HLH from other macrophage activation syndromes (MAS). For this study, we analyzed cases where MAS was ruled out after consultation with two co-authors who are immunologists. This is mentioned in the Materials and Methods section.

3. At what time point were presepsin samples obtained, before or after initiation of treatment?

This study included 61 consecutive cases with residual serum samples available for presepsin measurement following bone marrow aspiration. As shown in Table 1, the underlying diseases varied, and bone marrow aspiration was performed at a time point preceding therapeutic intervention for the underlying disease in most cases. The treatment was for the underlying disease, not for HLH. This is mentioned in the Materials and Methods section.

4. What is the mechanistic basis for elevated presepsin in HLH? The authors note that in sepsis, bacterial digestion of CD14 leads to its cleavage and subsequent cytokine release. In HLH, where red blood cells are phagocytosed, how does CD14 shedding occur from the macrophage surface?

CD14 is localized on the cell membranes of monocytes, macrophages, and neutrophils. Arai Y, et al. have reported that membrane form CD14 is internalized not only during bacterial phagocytosis but also during aseptic phagocytosis, such as that of sodium urate crystals. It is then cleaved by proteases like elastase and released as the soluble CD14, presepsin [Arai Y, et al. J Infect Chemother. 2015; 21: 564-569.]. In this study, we demonstrated reduced CD14 expression on macrophages in a HLH case. We hypothesized that this reduction likely resulted from the same mechanism previously reported. Specifically, CD14 on the macrophage surface was internalized during phagocytosis of blood cells, cleaved by proteases, and consequently led to decreased the expression of CD14 on the cell surface. It is mentioned in the Discussion section.

5. What are the typical presepsin levels observed in sepsis, and how do they compare with those in HLH?

Thank you for pointing out an important point. According to Yaegashi et al., the presepsin level was 294.2 ± 121.4 pg/mL in healthy individuals, 817.9 ± 572.7 pg/mL in sepsis patients, and 1,992.9 ± 1,509.2 pg/mL in severe sepsis patients [Yaegashi Y, et al. J Infect Chemother. 2005; 11: 234-238.]. On the other hand, Nanno et al. reported that the median presepsin level at HLH onset in patients with hematological malignancies was 1,935 pg/mL (range 182–11,800 pg/mL) [Nanno S, et al. Intern Med. 2016; 55: 2173-2184.]. They suggested that HLH cases often showed elevated levels comparable to those in severe sepsis, indicating a significant degree of macrophage activation. In our study, the presepsin level in a case meeting the HLH diagnostic criteria [Imashuku S. Int J Hamatol. 1997; 66: 135-151.] was 1,895 pg/mL, comparable to previously reported values. While presepsin elevation in sepsis primarily results from phagocytosis of bacterial components by monocytes, macrophages, and neutrophils [Arai Y, et al. J Infect Chemother. 2015; 21: 564-569.], in HLH, it is thought to arise mainly from excessive macrophage activation and hemophagocytosis due to dysregulation of immune control associated with underlying conditions such as malignant lymphoma or autoimmune diseases [Nanno S, et al. Intern Med. 2016; 55: 2173-2184.]. Therefore, while presepsin elevation is observed in both sepsis and HLH, we speculate that in sepsis it may reflect the immune response to infection, whereas in HLH it may reflect pathological overactivation due to immune dysregulation. It is mentioned in the Discussion section.

RESPONSE TO REVIEWER #2

Abstract

1. Page 5, line 2: background is more preferable than objective, but include your objective under the heading of background.

We have changed “Objective” to “Background”.

Introduction

1. Page 7, Line 16: space between the word “blood” and the reference.

We have addressed the point the reviewer mentioned.

Methods and materials

1. Participants is more formal than patients if possible.

The subtitle has been changed from “Patients” to “Participants.”

2. How many of your study participants are with HLH?

Thank you for your comment. In this study, 61 consecutive patients who underwent both bone marrow aspiration and serum presepsin measurement at the same time were included, not all cases were diagnosed with HLH. Among them, only one case was finally diagnosed with HLH, and the remaining 60 cases did not meet the diagnostic criteria for HLH [Imashuku S. Int J Hamatol. 1997; 66: 135-151.]. Supplementary Table 1 presents the frequency of findings based on the HLH diagnostic criteria analyzed in this study. Furthermore, we included a description of one case meeting the diagnostic criteria in the Materials and Methods and the Results section.

3. Page 9, line 4-5: Patients with an estimated glomerular filtration rate (eGFR) <30 (mL/min/1.73 m2) were excluded……. Needs citation (reference).

We apologize for the failing to include the reference. The reference [Nagata T, et al., PLoS One. 2015; 10: e0129159 ] has been added to support this exclusion criterion in this study.

4. Page 9, line 14: Serum samples were collected from residual patient samples. What was the tube used to collect the blood, at what centrifugal force was the blood centrifuged to separate the serum? Please state this section in detail.

We are sorry that the detailed is missing from the description. Serum samples were collected from residual patient samples. Those were obtained using coagulation-promoting tubes manufactured by Sekisui Medical Co., Ltd. (Tokyo, Japan). Peripheral blood samples were centrifuged at 3,500 rpm for 8 minutes, and the separated serum was stored at -80 °C until analysis. We have added the description in the Materials and Methods section;

5. Bone marrow samples were obtained from patients with B-cell lymphoma without bone marrow infiltration and from patients with HLH after bone marrow transplantation. Lymph node samples were collected from patients with lung cancer or B cell lymphoma. Why you have collected bone marrow and lymph node samples only from those study participants since there are other hematological and autoimmune diseases included in your result section (table 1)? State clearly please?

We apologize for the insufficient explanation. First, regarding Figure 1, since obtaining normal bone marrow specimens is difficult, we considered bone marrow specimens from a B-cell lymphoma case without malignant cell infiltration to be the most appropriate normal control. We assumed this bone marrow specimen to be the normal control. Next, regarding Figure 3, the bone marrow is from a patient with acute myeloid leukemia. This was the only HLH case in this study and was selected as a typical example (Fig. 3 j-l). The normal control for bone marrow was the same as above (Fig. 3 g-i). Finally, regarding lymph nodes, obtaining normal lymph nodes was also difficult. Therefore, lymph nodes without malignant cell metastasis from surgical specimens of a lung cancer patient (non-hematologic malignancy) were used as the normal control (Fig. 3 a-c). Meanwhile, the aforementioned HLH case with acute myeloid leukemia did not have lymph nodes collected. However, in the malignant lymphoma case in this study, lymph nodes were collected to evaluate the presence of hemophagocytosis. Since hemophagocytosis was observed, this specimen was adopted (Fig. 3 d-f).

6. You know that in histopathology lab, there are certain processes that the tissue section should undergo before mounting including tissue processing and staining. Thus, more detailed information is needed.

We are sorry that the detailed is missing from the description. We have revised the text to include detailed descriptions of tissue fixation, embedding, sectioning, and staining;

Bone marrow and lymph node samples from the subject patients were fixed in 10% neutral-buffered formalin, dehydrated through a graded ethanol series, and embedded in paraffin. Paraffin-embedded tissues were cut into 3-μm sections and mounted onto glass slides. The sections were deparaffinized in xylene, rehydrated through graded alcohols, and then subjected to hematoxylin and eosin (HE) staining or immunohistochemistry (IHC). For IHC, antigen retrieval was performed in Tris-EDTA buffer (pH 9.0) at 95 °C for 40 minutes. After blocking endogenous peroxidase with 3% H2O2 for 5 minutes, the slides were incubated with the primary antibodies CD14 (clone D7A2T, 1:200 dilution; Cell Signaling Technologies, MA, USA) and CD68 (clone PG-M1, 1:100 dilution; Dako, Glostrup, Denmark) for 30 minutes. Subsequently, the sections were incubated with visualization reagent (Histofine simple stain MAX-PO (MULTI); Nichirei Co., Tokyo, Japan) for 30 minutes, and the immunoreaction was visualized using the Dako REAL EnVision Detection System, Peroxidase/DAB+, Rabbit/Mouse (Dako) and counterstained with hematoxylin. We have described the above content in the Materials and Methods section.

7. Page 11, Line 10: Data are expressed as mean ± standard deviation (SD). Are all your data continuous and normally distributed? Additionally, were is more preferable than are.

Thank you for this very important comment. As the reviewer pointed out, not all data are normally distributed. Therefore, we have presented all data using both mean ± standard deviation (SD) and median (interquartile range: IQR), and changed to a nonparametric method using Spearman's rank correlation coefficient. We sincerely apologize, but since we had used Spearman's rank correlation coefficient for the analysis from the beginning, the results remain unchanged. Accordingly, we have revised the data presentation in the Materials and Methods section and Tables. We have also changed “are” to “were”.

Result

1. Result: Page 1, Line 10-12: LD, Ferritin and presepsin are singular. Thus, say level, not levels and was, not were or you can state as “The LD, Ferritin and presepsin levels were increased with their respective concentration of …….

We have addressed the point the reviewer mentioned.

2. Page 12, Line 14: D-dimer levels were elevated at 6.0 ± 9.0 μg/mL, as well as soluble fibrin (15.5 15 ± 23.6 μg/mL). Rewrite it like D-dimer as well as soluble fibrin levels were elevated (6.0 ± 9.0 μg/mL and 15.5 ± 23.6 μg/mL, respectively).

We have addressed the point the reviewer mentioned.

3. Page 12, Line 15-16: Bone marrow findings showed an increased nucleated cell count (77.8 ± 16 182.3×10⁹/L) and an elevated percentage of hemophagocytes (7.5 ± 12.7%). Define elevated nucleated cell and pathological percentage of hemophagocytes in bone marrow (i.e. need operational definitions).

Thank you for the valuable comment. The reference range for total nucleated cell count in human bone marrow aspirate was initially proposed by Yokomatsu Y, et al. as 10–25×10⁴/μL based on analysis of 236 aspirates [Yokomatsu Y, et al. Osaka City Med J. 1993; 39: 93-119.]. These reference values have since been adopted in studies evaluating automated counting of bone marrow aspirate. Most recently, this includes a report by Tsuchiya K, et al. using the Sysmex XN-3000 analyzer [Tsuchiya K, et al. Int J Lab Hematol. 2023;45:460-468.]. Therefore, a total nucleated cell count within the range of 10–25×10⁴/µL can reasonably be considered within the normal range.

As pointed out by the reviewers, there is no clear definition for the increased hemophagocytes or the pathological percentage of hemophagocytes within nucleated cells in the bone marrow. Previous studies have emphasized that myeloid hemophagocytic syndrome can be observed even in non-HLH settings, and that this finding alone has neither high sensitivity nor specificity [Goel S, et al. Ann Clin Lab Sci. 2012; 43: 21-25.]. Furthermore, standard hematology textbooks describe bone marrow phagocytosis not as a quantitative diagnostic criterion but as a supportive morphological feature [Greer JP, et al. Wintrobe's Clinical Hematology 11th ed.]. To clearly define a level significantly exceeding the sporadic or incidental hemophagocytosis seen in these reported cases, we operationally defined an increase in hemophagocytes as a state where hemophagocytes increase to 1 % or more of nucleated cells (i.e., when 5 or more hemophagocytes are observed per 500 counted nucleated cells).

We have described the above in the Materials and Methods section.

4. Page 13, Line 3: In 61 patients, the median serum presepsin level was 311.2 pg/mL (range 91-1895 pg/mL). It is preferred to say “in this study, the median serum presepsin level was 311.2 pg/mL (range 91-1895 4 pg/mL). Additionally, in your method and material section under” Statistical analysis” you have stated that “Data are expressed as mean ± standard deviation (SD)”. Here you stated the serum presepsin level in median (range), but in mean± SD (303.8 ± 288.8) as shown in table 2 and page 12, line 12 (check your analysis again please).

Thank you for pointing out an important point. We apologize for the confusion caused by inconsistent reporting of values as either mean or median. We have presented all data using both mean ± standard deviation (SD) and median (interquartile range: IQR). Accordingly, we have revised the data presentation in the Results section and Tables.

5. Page 13, Line 10: Additionally, presepsin level was significantly correlated with bone marrow findings, including……, Not Additionally, presepsin levels significantly correlated with bone marrow findings, including……

We have addressed the point the reviewer mentioned.

6. Page13, Line 15-16: Presepsin precursor CD14 protein expression is attenuated in hemophagoc

---

## [Decision Letter · Decision Letter 1]

14 Jan 2026

Dear Dr. Furuhashi,

Thank you for submitting your manuscript to PLOS ONE. After careful consideration, we feel that it has merit but does not fully meet PLOS ONE’s publication criteria as it currently stands. Therefore, we invite you to submit a revised version of the manuscript that addresses the points raised during the review process.

We look forward to receiving your revised manuscript.

Kind regards,

Zhaoqing Du, Ph.D

Academic Editor

PLOS One

Journal Requirements:

Reviewers' comments:

Reviewer's Responses to Questions

**Comments to the Author**

Reviewer #1: All comments have been addressed

Reviewer #2: (No Response)

2. Is the manuscript technically sound, and do the data support the conclusions?

Reviewer #1: Yes

Reviewer #2: Yes

3. Has the statistical analysis been performed appropriately and rigorously?

Reviewer #1: Yes

Reviewer #2: (No Response)

4. Have the authors made all data underlying the findings in their manuscript fully available?

Reviewer #1: Yes

Reviewer #2: Yes

5. Is the manuscript presented in an intelligible fashion and written in standard English?

Reviewer #1: Yes

Reviewer #2: Yes

Reviewer #1: Authors have addressed my concerns adequately and their responses are reflected in the revised manuscript.

Reviewer #2: PONE-D-25-46766R1

Title: Serum presepsin levels reflect macrophage activation and hemophagocytosis in bone marrow

Comments to the Author

Dear Authors,

Thank you for your revisions. The manuscript has been significantly improved as a result of your efforts. That said, I still have a few remaining minor comments that require your attention.

1. Page 14 line 8: Re-write it. “In this study, the serum presepsin level was mean 303.8 ± 288.8 pg/mL, and median 222 pg/mL (IQR 154-313 pg/mL).” Like as “In this study, the mean serum presepsin level was 303.8 ± 288.8 pg/mL, and that of its median was 222 pg/mL (IQR 154-313 pg/mL)”.

2. Please ensure proper use of subject-verb agreement throughout your document. For instance, page 17 line 1-4 “Presepsin level in sepsis patients, which are also commonly used in clinical practice, have been reported as 817.9 ± 572.7 pg/mL in sepsis patients and 1,992.9 ± 1,509.2 pg/mL in severe sepsis patients, compared to 294.2 ± 121.4 pg/mL in healthy individuals”. This statement refers to Presepsin level, not septic patients. Thus, are should be replaced by is and have been by has been, similarly line 10… Elevated presepin level in sepsis are primarily attributed….

Best regards,

**Do you want your identity to be public for this peer review?** For information about this choice, including consent withdrawal, please see our Privacy Policy

Reviewer #1: **Yes:** Vahid Afshar-Kharghan

Reviewer #2: No

---

## [Author Response · Author response to Decision Letter 2]

15 Jan 2026

RESPONSE TO REVIEWER #2

1. Page 14 line 8: Re-write it. “In this study, the serum presepsin level was mean 303.8 ± 288.8 pg/mL, and median 222 pg/mL (IQR 154-313 pg/mL).” Like as “In this study, the mean serum presepsin level was 303.8 ± 288.8 pg/mL, and that of its median was 222 pg/mL (IQR 154-313 pg/mL)”.

We have addressed the point the reviewer mentioned.

2. Please ensure proper use of subject-verb agreement throughout your document. For instance, page 17 line 1-4 “Presepsin level in sepsis patients, which are also commonly used in clinical practice, have been reported as 817.9 ± 572.7 pg/mL in sepsis patients and 1,992.9 ± 1,509.2 pg/mL in severe sepsis patients, compared to 294.2 ± 121.4 pg/mL in healthy individuals”. This statement refers to Presepsin level, not septic patients. Thus, are should be replaced by is and have been by has been, similarly line 10… Elevated presepin level in sepsis are primarily attributed….

We apologize for the incorrect description. We have addressed the point the reviewer mentioned.

---

## [Decision Letter · Decision Letter 2]

27 Feb 2026

Serum presepsin level reflects macrophage activation and hemophagocytosis in bone marrow

PONE-D-25-46766R2

Dear Dr. Furuhashi,

We’re pleased to inform you that your manuscript has been judged scientifically suitable for publication and will be formally accepted for publication once it meets all outstanding technical requirements.

Kind regards,

Yoshito Nishimura, MD, PhD, MPH

Academic Editor

PLOS One

Additional Editor Comments (optional):

Reviewers' comments:

Reviewer's Responses to Questions

**Comments to the Author**

Reviewer #1: All comments have been addressed

Reviewer #2: All comments have been addressed

2. Is the manuscript technically sound, and do the data support the conclusions?

Reviewer #1: Yes

Reviewer #2: Yes

3. Has the statistical analysis been performed appropriately and rigorously?

Reviewer #1: Yes

Reviewer #2: Yes

4. Have the authors made all data underlying the findings in their manuscript fully available?

Reviewer #1: Yes

Reviewer #2: Yes

5. Is the manuscript presented in an intelligible fashion and written in standard English?

Reviewer #1: Yes

Reviewer #2: Yes

Reviewer #1: I have no additional comments or suggestions. Authors had addressed my concerns in the previous revisions.

Reviewer #2: PONE-D-25-46766R2

Title: Serum presepsin levels reflect macrophage activation and hemophagocytosis in bone marrow

Comments to the Author

No comments.

**Do you want your identity to be public for this peer review?** For information about this choice, including consent withdrawal, please see our Privacy Policy

Reviewer #1: No

Reviewer #2: No

---

## [Editor Report · Acceptance letter]

PONE-D-25-46766R2

PLOS One

Dear Dr. Furuhashi,

I'm pleased to inform you that your manuscript has been deemed suitable for publication in PLOS One. Congratulations! Your manuscript is now being handed over to our production team.

Kind regards,

on behalf of

Dr. Yoshito Nishimura

Academic Editor

PLOS One